# Differential impact of isolated topographic bumps on ice sheet flow and subglacial processes

Marion A. McKenzie[1], Lauren E. Miller[1], Jacob S. Slawson[1*], Emma J. MacKie[2], and Shujie Wang[3]

[1]Department of Environmental Sciences, University of Virginia, 291 McCormick Rd., Charlottesville, VA, USA 22904

[2]Department of Geological Sciences, University of Florida, 241 Williamson Hall, Gainesville, FL, USA 32611-2120

[3]Department of Geography, Pennsylvania State University, 302 N Burrowes St., University Park, PA, USA 16802

*Current affiliation: Department of Geology and Geological Engineering, Colorado School of Mines, 1516 Illinois St., Golden, CO, USA 80401

*Correspondence to*: Marion A. McKenzie (mm8dt@virginia.edu)

**Abstract.** Topographic highs ("bumps") across glaciated landscapes have the potential to temporarily slow ice sheet flow or, conversely, accelerate ice flow through subglacial strain heating and meltwater production. Isolated bumps of variable size across the deglaciated landscape of the Cordilleran Ice Sheet (CIS) of Washington state present an opportunity to study the influence of topographic highs on ice-bed interactions and ice flow organization. This work utilizes semi-automatic mapping

techniques of subglacial bedforms to characterize the morphology of streamlined subglacial bedforms including elongation, surface relief, and orientation, all of which provide insight into subglacial processes during post-Last Glacial Maximum deglaciation. We identify a bump-size threshold of several cubic kilometers -- around 4.5 km$^3$ -- in which bumps larger than this size will consistently and significantly disrupt both ice-flow organization and subglacial sedimentary processes which are fundamental to the genesis of streamlined subglacial bedforms. Additionally, sedimentary processes are persistent and well-

developed downstream of bumps as reflected by enhanced bedform elongation and reduced surface relief, likely due to increased availability and production of subglacial sediment and meltwater. While isolated topography plays a role in disrupting ice flow, larger bumps have a greater disruption to ice flow organization, while bumps below the identified threshold seem to have little effect on ice and subglacial processes. The variable influence of isolated topographic bumps on ice flow of the CIS has significant implications for outlet glaciers of the Greenland Ice Sheet (GrIS) due to similarities in regional

topography where local bumps are largely unresolved.

## 1 Introduction

Isolated topographic highs in the terrain beneath ice sheets can contribute to increased basal drag and decreased ice flow velocity and, for marine-based margins, offer pinning-points to halt or slow down margin retreat (Durand et al., 2011; Favier

et al., 2016; Alley et al., 2021; Robel et al., 2022). Conversely, ice flow over topographic highs can increase strain heating and basal meltwater production, elevating basal meltwater pressure and reducing basal friction in the downstream environment (Payne and Dongelmans, 1997; Cuffey and Paterson, 2010). However, identifying which forms and scales of "bumps" across a glaciated landscape may increase, decrease, or not affect ice-flow velocity, basal water pressure, and basal friction is not well understood outside of simple geophysical modeling (Alley et al., 2021). Additionally, topography at the base of the ice sheet, and even for most glacier catchments, is poorly resolved (e.g., MacKie et al., 2020; Morlighem et al., 2020). Therefore, we turn to a formerly glaciated landscape in Washington state where geomorphological indicators of ice-flow conditions in the form of streamlined subglacial bedforms (e.g., glacial lineations, whalebacks, and drumlins) can be used to better understand the sensitivity of ice sheets to isolated bumps in the subglacial environment. Due to the controls of basal shear, meltwater and sediment availability, and sediment processes including transport, deposition, and erosion on bedform synthesis, morphometrics of streamlined subglacial bedforms offer information on ice-bed interactions (Schoof and Clark, 2008; Shaw et al., 2008; King et al., 2009). Quantitatively assessing streamlined subglacial bedform morphometrics provides an opportunity to assess characteristics of paleo-ice flow organization and relative speeds across a landscape (e.g., Clark, 1997, 1999; King et al., 2009; Clark et al., 2003, 2009; Spagnolo et al., 2012, 2014; Principato et al., 2016). Assessment of streamlined subglacial bedforms and their implications for ice flow are applicable to modern ice sheets (MacKie et al., 2021), where observations and theory of subglacial conditions are spatially and temporally limited.

## 1.1 Site Characteristics

The Puget Lowland of Washington state was glaciated by the southwestern Cordilleran Ice Sheet (CIS) during the Last Glacial Maximum (LGM), when the region was largely depressed below sea level due to glacial isostatic adjustment (GIA; Booth and Hallet, 1993; Dethier et al., 1995; Kovanen and Slaymaker, 2004; Eyles et al., 2018); therefore, the southwestern CIS was predominantly marine based. The Puget Lowland was glaciated at least six times by the CIS during the Quaternary (Clague & James, 2002; Booth et al., 2003) indicating the highly active growth and decay of the marine-terminating southern lobe. The drivers of CIS behavior are not well understood, as the growth and decay of this formidable-size ice sheet has not been found to coincide with global perturbations like its eastern neighbor, the Laurentide Ice Sheet (Broecker, 1994; Cheng et al., 2016; Blunier & Brook, 2001; Walczak et al., 2020). Active tectonics and volcanic activity across the Puget Lowland led to exposed crystalline and volcanic bedrock of Eocene age interrupting sedimentary bedrock across the region (Fig. 1A; Khazaradze et al., 1999; Booth et al., 2004). On a larger scale, the Puget Lowland is a basin surrounded by mountainous terrain near the coast of the Pacific Ocean with isolated topographic highs, similar to the terrain beneath the margins of the Greenland Ice Sheet (Fig. 1; Bamber et al. 2013; Eyles et al., 2018). The variability in the hard to soft-bed conditions in the Puget Lowland also bears similarity to the mixed-bed conditions seen beneath Thwaites Glacier (Schroeder et al., 2014; Holschuh et al., 2020; Alley et al., 2021). Based on simple geophysical subglacial models focused on Thwaites Glacier (Alley et al., 2021), it is likely high relief regions influence marine-based ice-bed interactions through basal drag and sediment reorganization. As such, using

geomorphic evidence across the crystalline and volcanic bedrock exposures or "bumps" in the Puget Lowland likely elucidates the influence of isolated topography on ice flow and sedimentary processes in a subglacial environment, but this concept has yet to be empirically tested across the region. This work aims to determine the role of topographic bumps on glacial ice flow and sedimentary processes via streamlined subglacial bedform morphology and distribution due to its relevance for

constraining the influence of subglacial topography beneath contemporary glacial ice. By investigating ice flow behavior within a single glacial system, the effects of isolated crystalline bedrock highs on ice flow will not be confounded by geographically variable conditions such as local climate and ocean forcings. This work is also not confounded by the different timescales of deglaciation and post-glacial landscape evolution due to similar deglaciation and post-glacial landscape evolution processes across the region.

**2. Methodology**

**2.1 Topographic "Bump" Classification**

Digital elevation models (DEMs) with horizontal resolution of 1.83 x 1.83 meters and vertical resolution of 2 meters from across the Puget Lowland (Clallam County, Olympic Department of Natural Resources, WA, 2008; Quantum Spatial Inc., 2017, 2019; OCM Partners, 2019a, 2019b) and ambient occlusion hillshading techniques (c.f., McKenzie et al., 2022) were

used to analyze nine crystalline and volcanic bedrock bumps for streamlined subglacial bedforms across the Puget Lowland. The nine identified bumps span a wide range in peak elevation, bump surface area, and bump volume (Fig. 1). While some bumps have more coarse topography than others, the lack of ice streaming seen in-between isolated topographic highs with a singular 100-foot contour outline, as determined by the presence of streamlined subglacial bedforms, allowed us to treat all bumps as "aggregate" features. The outermost 100-foot closed contour across each aggregate bump was expanded to three

times the surface area to classify the region of interest, following the modeled influence of bump perturbations on basal hydrologic potential (Alley et al., 2021). While present-day elevations of these deglaciated sites differ from elevations during glaciation due to GIA, tectonics, and post-glacial landscape evolution, the scale of bump relief in relation to streamlined bedforms is well preserved. Fractures, faults, and joints from tectonic activity and brittle deformation of the crust across bumps are below the scale of analysis for this work and are therefore not considered here.


**2.2 Streamlined subglacial bedform identification**

Streamlined subglacial bedforms were identified across the nine bump sites using a combination of Topographic Position Index (TPI) analysis (McKenzie et al., 2022), contour-tree mapping (Wang et al., 2017), and manual identification. TPI utilizes DEM slope variations across defined cell-neighborhood sizes to semi-automatically identify positive relief features (McKenzie et

al., 2022). Morphometric threshold limits were applied to TPI-mapped bedform length, width, and area to increase the accuracy of the tool. In the larger Puget Lowland region, the TPI tool generally maps 80% of the bedforms correctly (McKenzie et al., 2022). Additionally, localized contour-tree mapping was utilized on the DEM data to isolate closed contours within a defined

elevation (Wang et al., 2017). Both tool outputs were validated and corrected through manual removal of incorrectly identified features and manual addition of bedforms missed by the automated tools, resulting in a highly accurate dataset of bedforms

across the region (McKenzie et al., 2022).

All bedforms in the final dataset (n=3,273) have an associated long-axis length, cardinal orientation, width orthogonal to long-axis length, and range in elevation across the long axis. These metrics were calculated by the ArcGIS Pro "Minimum Bounding Geometry" and "Add Z Information" tools (McKenzie et al., 2023). Long axis cardinal direction, or orientation, of streamlined bedforms is used to infer direction of ice flow (Clark, 1997; Kleman et al., 2006; Kleman and Borgström, 1996).

Bedform elongation ratio, calculated by dividing a bedform's length by its width is used to infer relative speed of ice flow velocity (Clark 1997, 1999; Clark et al., 2003) and relative duration of ice presence in a region (Benediktsson et al., 2016). Bedform surface relief, the difference between the highest and lowest elevation along the bedform long axis, is used to infer persistence of ice flow and sedimentary processes in the subglacial environment (McKenzie et al., 2022). Smaller surface relief values indicate more persistent, warm-based ice flow across the bed with well-developed sedimentary processes in the

subglacial environment than larger values of bedform surface relief (McKenzie et al., 2022). The spatially uniform rate of GIA in the region (centimeters per year; Dethier et al., 1995), is much larger than rates of regional tectonics that also differ spatially (millimeters per year; Sherrod et al., 2008). Therefore, along the long axis of a single bedform, impacts of post-glacial erosion and regional-scale tectonics have spatially variable influence and cannot be isolated from larger-scale dynamics such as GIA and glacial sedimentary processes. Due to these spatial discrepancies, a high resolution assessment of where bedform surface

relief occurs across an individual long axis was not conducted in this work.

For each site, bedforms were categorized into groups "upstream", "on top of", and "downstream" as determined by bedform location with respect to the outermost 100-foot closed contour of the topographic high. There is a potential for bumps to develop lateral shear margins as a result of increasing ice speed along the edge of the bump (Alley, 1993; Stokes et al., 2007). Therefore, bedforms mapped alongside the bump were classified as "downstream" of the bump, considering the

overlying ice in the lateral direction would have contacted the bump. While bedforms themselves cause shear stress on ice (Damsgaard et al., 2020), the scale of the bumps to bedforms makes the influence of bedforms on ice flow negligible in this case study. Additionally, uniform glaciation in this region over at least several hundred thousand years contributes to a consideration of these bedforms as a final "snap shot" of bedform development.

Bedforms in the upstream and downstream environments are likely to be sedimentary forms due to large sediment

availability, while bedforms on top of bumps are interpreted to be erosional features developed in the sediment-starved environment of the crystalline bedrock highs. While the difference in generation is loosely referred to throughout this work, the specific sedimentary composition of the bedform was not explicitly considered as we analyze bedform metrics collectively for all erosional and depositional features. Defining bedforms solely on shape rather than composition is a strength of this work as it considers topographic influence on ice flow and sediment processes without commenting on specific landform-

generating processes (McKenzie et al., 2022).

To quantify bump influence, we performed analysis of variance (ANOVA) and non-parametric Kruskal-Wallis tests on groups of streamlined subglacial bedforms across bumps to compare the statistical significance of the means and distributions between populations, respectively, in "R". Results of statistical analyses were used to determine significance ($p <$ 0.05) of bedform characteristics at each site (i.e., upstream, on top of, and downstream of bumps) as well as significance of bedform morphometrics across sites, where significance is defined as a p value less than 0.05 between groups.

## 3. Results

The number of streamlined bedforms per site is positively correlated with bump surface area and volume (Fig. 1). When considering morphologies of streamlined bedforms across bumps, bedform elongation for the full dataset (n = 3,273) is lowest and bedform surface relief is highest on top of bumps, rather than upstream or downstream (Fig. 2). Differences between upstream and downstream subglacial streamlined bedform morphometrics are not notably different across many of the sites (Fig. 3), however, one difference of note is the increased number of bedforms downstream of bumps than upstream (Fig. 2). At seven of the nine sites, surface relief along bedform crests increases significantly between populations upstream and on top of bumps (average increase of 21 meters; Fig. 3B). The greatest number and most elongate bedforms, as well as the greatest proportion of bedforms with low surface relief, occur downstream of bumps (Fig. 1A; Fig. 2; Fig. 4A, 4B). Notably, there is a statistically significant decrease in bedform elongation between upstream and on top of the two largest bumps (average decrease of 1.5; Fig. 3A), Blue Hills and Devils Mountain.

While bedform surface relief and elongation ranges overlap across all site populations, bedforms associated with smaller bumps tend to have outliers below the 1$\sigma$ (68%) confidence level for all populations (e.g., San Juan Island, Fidalgo Island, and Black Hills) while those associated with larger bumps have outliers above the 1$\sigma$ confidence level (e.g., Blue Hills and Cougar Mountain; Fig. 2). The greatest proportion of bedforms with low surface relief (Fig. 4B) are located at the smallest bump sites ($< 0.3$ km$^3$). Many sites showcase an increase in disorganization of bedform orientation on top of the bump (Fig. 4C), but only at the two largest bumps ($> 4.5$ km$^3$) does downstream bedform orientation recover to patterns present in the upstream bedform populations (average orientation within 0.5 degrees similarity; Fig. 4C). The remaining sites have bedform orientations that either remain unchanged or develop less agreement in bedform orientation downstream (Fig. 4C).

## 4. Discussion and Interpretation

Positive correlation between site area and number of bedforms indicates spatial continuity in the bedform distribution across the Puget Lowland. Where a reduction in bedform elongation and an increase in surface relief occurs on top of bumps, this relationship suggests a slow in ice flow speed or persistence at the same time as a reduction in spatial homogeneity of sedimentary processes in developing streamlined subglacial bedforms. As such, we interpret that bumps in the subglacial environment of the CIS generally led to ice-flow deceleration and a reduction of spatial homogeneity (i.e., efficiency) of sedimentary processes including bedrock erosion and sediment transport and deposition – all of which are important for bedform genesis (Schoof and Clark, 2008; Shaw et al., 2008; King et al., 2009).

160        The effect of bumps on subglacial processes and ice flow are seen on top of almost all site bumps (Fig. 3). The nearly ubiquitous observation of bedform surface relief increase on top of bumps is most likely due to a transition in lithology from sedimentary to crystalline or volcanic bedrock, which disrupts along-flow sedimentary processes as ice begins developing subglacial streamlined bedforms on more-erosion-resistant and sediment-starved bed compositions. The two exceptions to this trend are Big Skidder Hill and Lopez Island, where there is no appreciable change in streamlined subglacial bedform surface

relief across the bumps (Fig. 3B). This suggests that the conditions at these two sites were able to overcome direct lithologic impact on bedform relief. Due to the more-erosion-resistant lithologies of the bumps, despite increased pressure and basal drag in the subglacial environment, we propose decreased efficiency in which the ice is able to facilitate streamlined subglacial bedform formation through bedrock erosion (Eyles and Doughty, 2016; Krabbendam et al., 2016), leading to truncated bedforms with high surface relief (McKenzie et al., 2022; Fig. 2). Despite the influence of bumps on ice flow and sedimentary

processes at almost all sites, the similarities between upstream and downstream bedform metrics are notable. However, there are far more streamlined subglacial bedforms identified downstream of bumps. Increased sediment availability and basal meltwater that results from the strain heating on top of the bump (Payne and Dongelmans, 1997) increases downstream sediment transport efficiency (McIntyre, 1985; Pohjola and Hedfors, 2003; Winsborrow et al., 2010b) which could contribute to development of a greater number of bedforms and recovery of the ice flow and sedimentary systems to reflect bedform

metrics seen upstream, prior to bump influence.

        Bumps with surface area generally below a few square kilometers and volume smaller than around 0.3 cubic kilometers have predominantly low-elongation, low-surface relief bedform outliers while large sites have highly elongate, high surface relief bedform outliers demonstrating a linkage between bump size and possible bedform morphometrics in a relatively systemic manner across the Puget Lowland (Fig. 2). We postulate that bump size – through its impact on ice flow and subglacial

processes – is a control on downstream bedform metrics. From observations of bedform orientation recovery only at the largest two sites, we infer that bump volume of over several cubic kilometers ($>4.5$ km$^3$) will support reorganization of downstream ice-flow orientation and subglacial sedimentary processes. Conversely, bumps below this several cubic kilometer threshold affect ice flow in such a way that the ice flow organization does not appear to regain the same organization seen upstream of bumps. There is little geomorphic evidence of channelized meltwater in the subglacial environment (Fig. 1A), suggesting that

meltwater development across these bumps was distributed and saturated in the porous upstream and downstream environments with water films supported atop the crystalline bedrock highs. An increase in subglacial meltwater downstream of bumps is supported by the presence of homogeneity in bedform metrics, which would support homogenous deformation of sediments into streamlined bedforms (Shaw et al., 2008; King et al., 2009).

        An additional consideration for this work is its relevance to modern subglacial systems with similar topographic

conditions. While local climatic factors render the specific thresholds for bump influence on ice and sedimentary process behavior relevant only in this glacial system, the process-based understanding derived here can be applied to systems where isolated topographic highs are identified. Topographic similarities between this region and outlet glaciers in Greenland (Eyles et al., 2018), modern Thwaites (Holschuh et al., 2020; Alley et al., 2021), and other contemporary glaciers with isolated

topographic highs (Greenwood et al., 2021) highlight the need for closer study of the effect of subglacial bumps on ice flow
in these systems with their own unique set of local climatic influences.

## 4. Conclusions

Overall, our results suggest there is general ice flow deceleration and reduction of bedrock erosion efficiency on top of bumps, which results from a subglacial lithology transition. Sedimentary processes, essential to streamlined bedform
genesis, are organized and efficient downstream of bumps despite disruption from crystalline bedrock highs - likely as a result of increased sediment availability and subglacial meltwater sourced from strain heating on top of the bump. The two largest bumps in both volume and surface area considered in this work notably disturb ice-flow orientation and speed on top of the bump. These same bumps are also the only that indicate recovery of ice flow orientation and speed downstream of bumps. Findings from these paleo-subglacial bumps may be used as an analog for ice flow in contemporary ice sheets and support
process-based understanding of subglacial terrain influence on overlying ice-sheet behavior in similar systems such as those beneath Thwaites Glacier and outlet glaciers within the Greenland Ice Sheet.

## 5. Data availability

All bedform data produced from this work is publicly available through PANGAEA and are available by request from the
corresponding author.

## 6. Author contribution

Project conceptualization, data curation, methodology, formal analysis, initial draft writing, and editing were conducted by M. McKenzie. Conceptualization, funding acquisition, formal analysis, editing, and supervision were conducted by L. Miller.
Conceptualization, preliminary research, and editing were conducted by J. Slawson. Partial conceptualization and editing were conducted by E. MacKie. Data curation support and editing were conducted by S. Wang.

## 7. Competing interests

The authors declare that they have no conflict of interest.

## 8. Acknowledgments

We acknowledge the Washington Department of Natural Resources for their accessible LiDAR data that made this project possible. The sites analyzed for this work are located on land historically cultivated and inhabited by the Skokomish, Suquamish, Squaxin, Stl'pulmsh, Steilacoom, Puyallup, Muckleshoot, and Duwamish peoples, while much of the data analysis
and interpretation were conducted on land cultivated and inhabited by the Monacan Nation. The peoples of these Nations were custodians of the land for time immemorial before forced removal and genocide during colonization. The authors acknowledge

their ongoing stewardship of the lands. This work was funded by the Chamberlain Endowment and the H.G. Goodell Endowment at the University of Virginia.

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

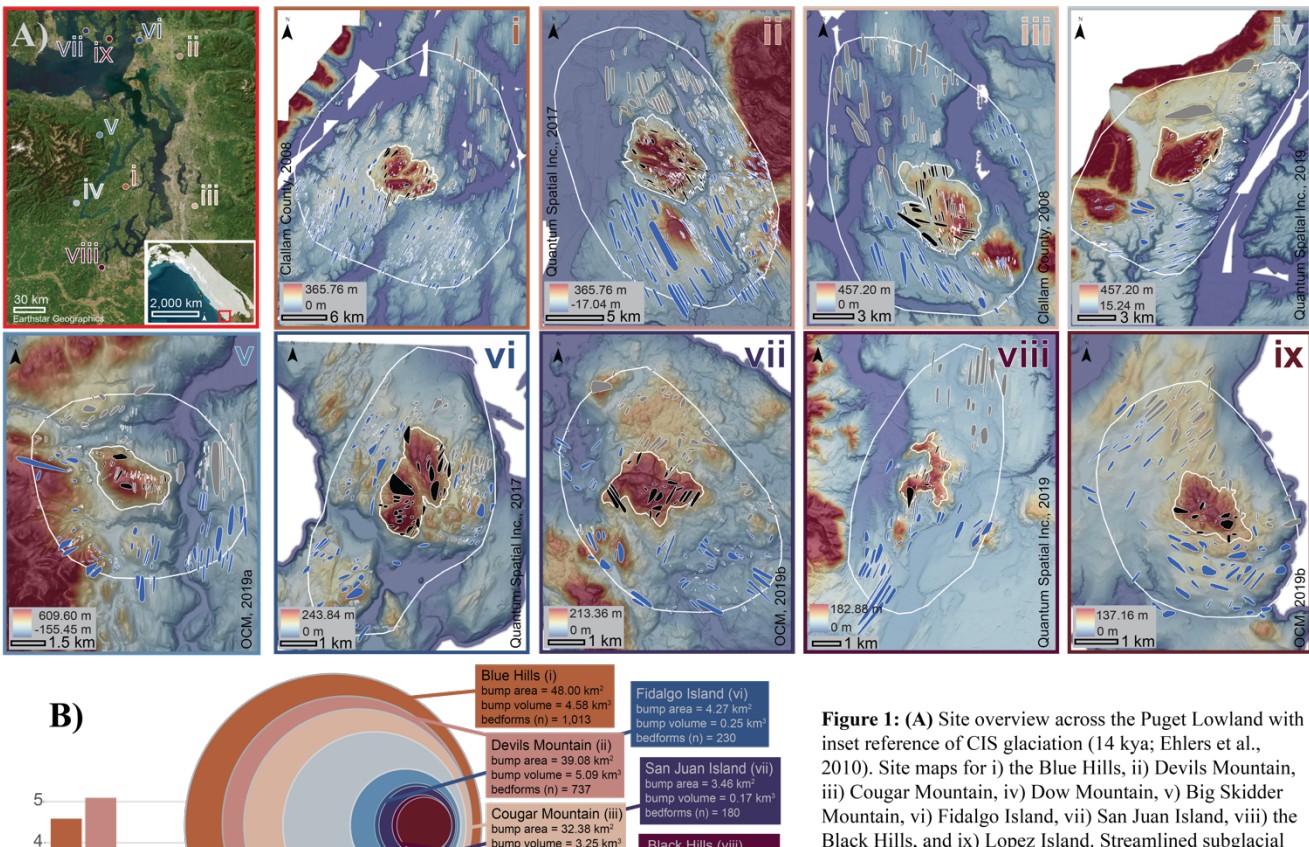

**Figure 1: (A)** Site overview across the Puget Lowland with inset reference of CIS glaciation (14 kya; Ehlers et al., 2010). Site maps for i) the Blue Hills, ii) Devils Mountain, iii) Cougar Mountain, iv) Dow Mountain, v) Big Skidder Mountain, vi) Fidalgo Island, vii) San Juan Island, viii) the Black Hills, and ix) Lopez Island. Streamlined subglacial bedforms are mapped upstream (gray polygons), on top of (black polygons), and downstream (blue polygons) of bed bumps (small white outlines) within larger site regions (large white outlines). **(B)** Relative volume and surface area of bed bump sites.

**Figure 1. A)** Site overview across the Puget Lowland with inset reference of CIS glaciation (14kya; Ehlers et al., 2010). Site maps for i) the Blue Hills, ii) Devils Mountain, iii) Cougar Mountain, iv) Dow Mountain, v) Big Skidder Mountain, vi) Fidalgo Island, vii) San Juan Island, viii) the Black Hills, and ix) Lopez Island. Streamlined subglacial bedforms are mapped upstream (gray polygons), on top of (black polygons), and downstream (blue polygons) of bed bumps (small white outlines) within larger site regions (large white outlines). **B)** Relative volume and surface area of bed bump sites.

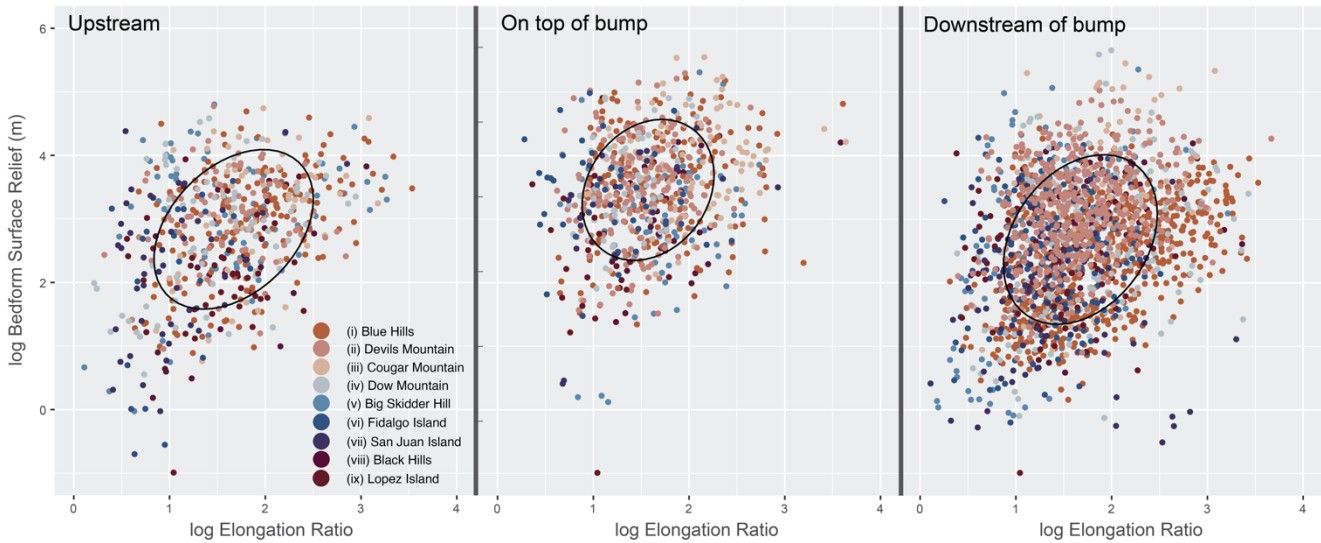

**Figure 2:** Scatterplots of the log of bedform elongation ratio and surface relief in meters. The ellipses are 1σ (68%) confidence levels for multivariate t-distributions for all bedforms (n=3,273). Sites are listed in the legend from largest surface area ((i) Blue Hills) to smallest surface area ((ix) Lopez Island).

**Figure 2. Scatterplots of the log of bedform elongation ratio and surface relief in meters. The ellipses are 1σ (68%) confidence levels**
**for multivariate t-distributions for all bedforms (n=3,273). Sites are listed in the legend from largest surface area ((i) Blue Hills) to smallest surface area ((ix) Lopez Island).**

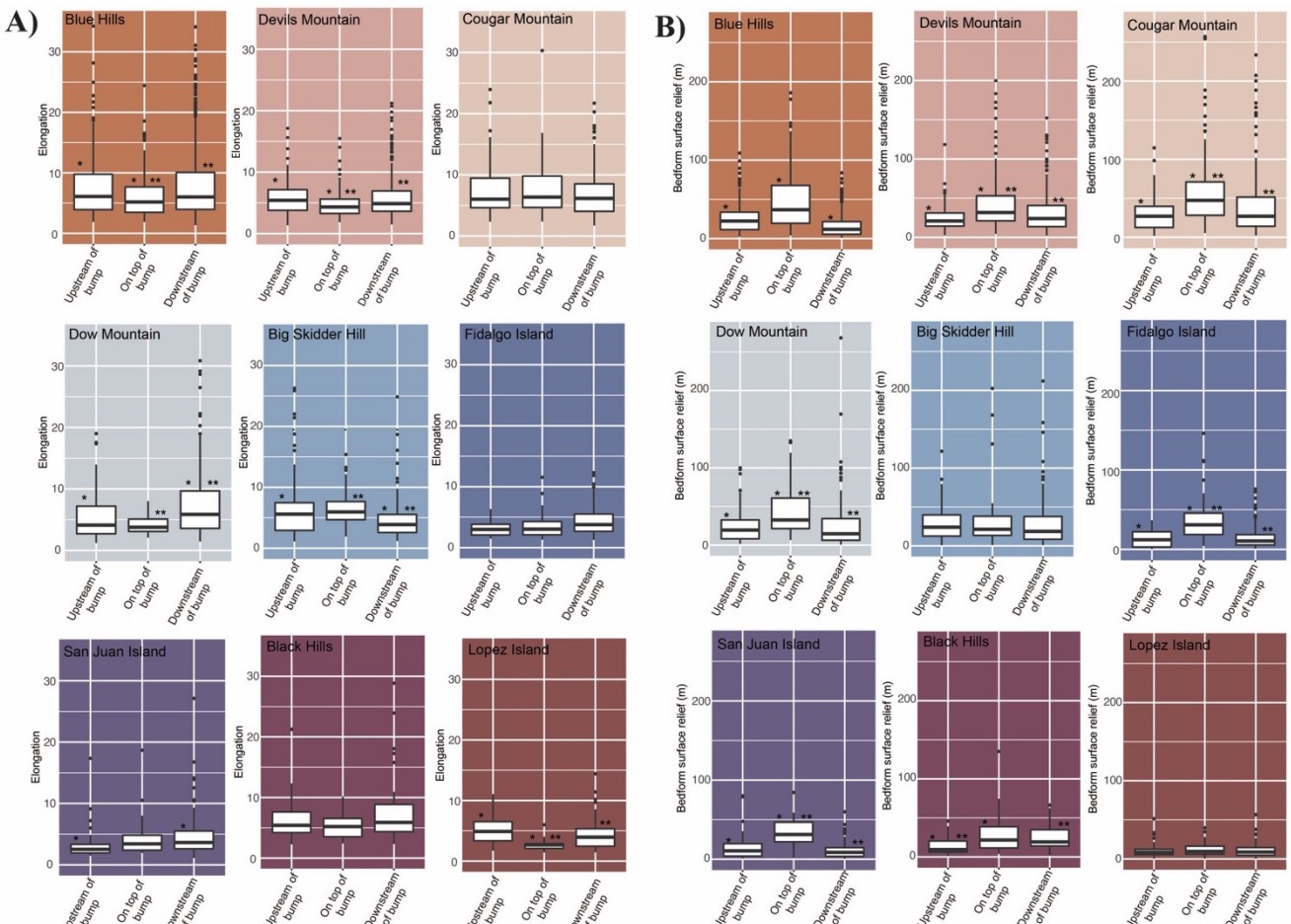

**Figure 3**: Box plots of A) bedform elongation data at each site with populations characterized upstream of, on top of, and downstream of bumps and B) bedform surface relief data at each site with populations characterized upstream of, on top of, and downstream of bumps. Statistically signficant differences between groups are indicated by asterisks. Multiple asterisks indicate a separate population with signficant differences, independent from other groups of statistical significance.

**Figure 3. Box plots of A) bedform elongation data at each site with populations characterized upstream of, on top of, and downstream of bumps and B) surface relief data at each site with populations characterized upstream of, on top of, and downstream of bumps. Statistically significant differences between groups are indicated by asterisks. Multiple asterisks indicate a separate population with significant differences, independent from other groups of statistical significance.**


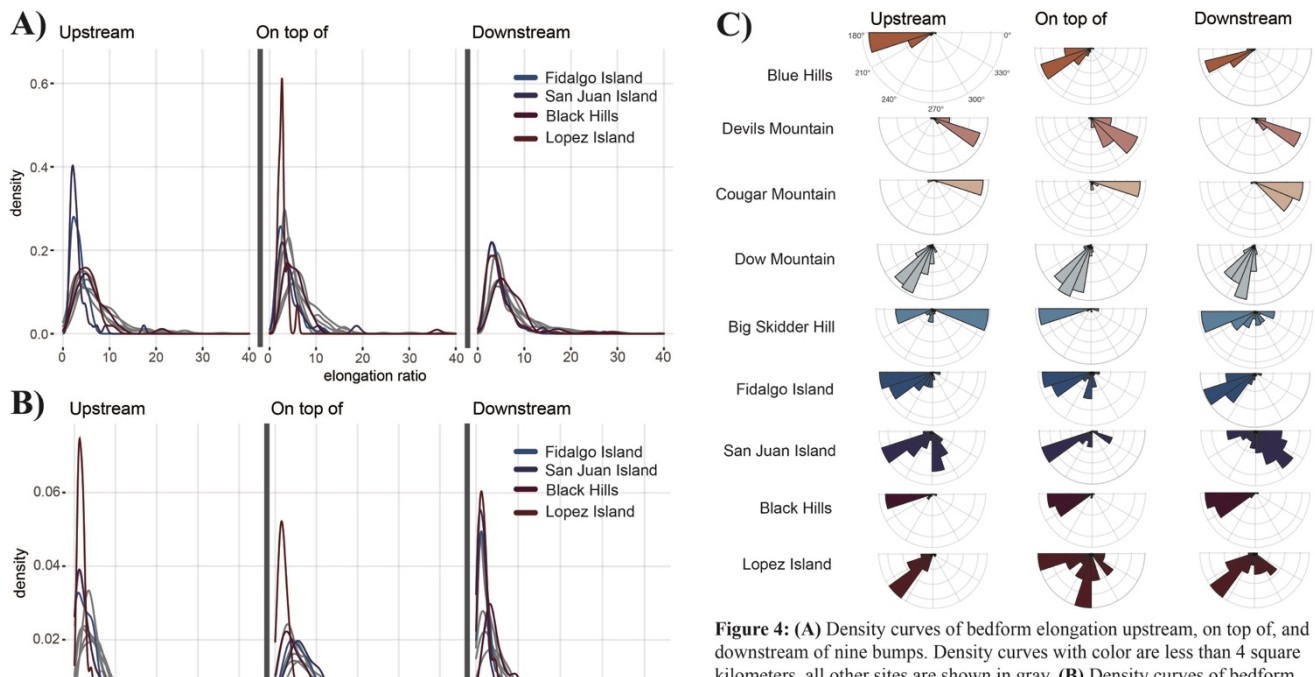

**Figure 4: (A)** Density curves of bedform elongation upstream, on top of, and downstream of nine bumps. Density curves with color are less than 4 square kilometers, all other sites are shown in gray. **(B)** Density curves of bedform surface relief in meters upstream, on top of, and downstream of nine bumps. Density curves with color are less than 4 square kilometers in surface area, all other sites are shown in gray. **(C)** Cardinal orientations of streamlined bedforms upstream, on top of, and downstream of nine bumps.

**Figure 4. A)** Density curves of bedform elongation upstream, on top of, and downstream of nine bumps. Density curves with color are less than 4 square kilometers, all other sites are shown in gray. **B)** Density curves of bedform surface relief in meters upstream, on top of, and downstream of nine bumps. Density curves with color are less than 4 square kilometers in surface area, all other sites are shown in gray. **C)** Cardinal orientations of streamlined bedforms upstream, on top of, and downstream of nine bumps.