# Peer review of "Differential impact of isolated topographic bumps on ice sheet flow and subglacial processes"

_EGUsphere, 2022_

## Author Response (AR1)

**REVIEW 1**

**Thank you for your comments and time spent assessing this work. All of your specific comments and suggestions are addressed in the following text.**

Specific comments:

Methods: The vertical and horizontal resolution of the DEMs used as the basis for bedform mapping needs to be added.

**This information was added in line 56 of the manuscript: "Digital elevation models with horizontal resolution of 1.83 x 1.83 meters and coarsest vertical resolution of 2 meters from across the Puget Lowland […]". Because many of the DEMs used to ensure full coverage of these sites had varying resolution, only the coarsest resolution was explicitly stated.**

Figure 1: The colour choices in this figure for both the background elevation data and the mapping mean that there is little contrast between the two, and the mapping is hard to see (especially the green bedforms appearing on the lowest topography. I suggest modifying the colour schemes to improve this. The elevation scale for each of the panels in A) also varies. In order to compare across sites it would be preferable to use a single colour scale for elevation in each panel. There should also be a space between the value and the unit in the labels. The study area panel is very dark. For those unfamiliar with the region there should be inclusion of a small inset to show the wider context of the location within the area covered by the Cordilleran Ice Sheet. The area comparisons in (B) is a very useful figure, but would be easier to follow if the lines connecting the labels to the graduated circles were thinner, and the colours of the text boxes were the same as the circles, especially as this same colour scheme is continued throughout the other figures to identify each site. It is unclear what the volume values are pertaining to in the labels. Is this of the area, the bump, or the bedforms?

**The color of the upstream bedforms was modified to gray. All bedforms were highlighted with a white outline to make them more visible.**

**Due to the large scale elevation differences between the bump sites themselves, one elevation scale was not utilized, but rather the same color range was kept constant between sites. In order to see variability in elevation across each site, the individual elevation ranges were needed.**

**A space was added between the elevation values and units in all labels.**

**The study area panel was updated to be brighter and more visible. In this panel, I also added an inset map of the Cordilleran Ice Sheet glaciation with an outline of the Puget Lowland.**

**Fig 1B was updated by reducing the size of the lines connecting the labels to the circles and the color of the text boxes was changed to match the circles. A bar graph was created to visually represent the bump volume. Additional text clarification of size references for each bump was added to text boxes in Figure 1B.**

**All relevant changes to the figure were changed accordingly in the Figure 1 caption.**

Figure 2: Include in the legend the site codes as used in Fig 1 as well as the names, or add site names to the labelling in Figure 1A.

**The site codes used in Fig 1 were added to the legend in Fig 2 and also added to the legend.**

I would recommend acceptance of the paper with corrections as specified under specific comments.

**REVIEW 2**

**Thank you for these comments. Your consideration of our logic in downstream ice-flow velocity is very helpful and will strengthen this work. All of your comments have been addressed in the following text.**

The manuscript is exceptionally well written, and I do not have many suggestions for changes. I suggest publication after the following points have been considered:

1.      L27-29: *"Conversely, ice flow over topographic highs can increase strain heating and basal meltwater production, thereby increasing ice-flow velocity downstream of the obstacle."*

I agree with the logic that increased strain heating leads to bed weakening downstream

of the bump through elevated basal water pressure and decreased effective stress. However, despite the basal weakening, the bump may not cause faster ice flow, as friction from the bump is transferred through the ice column by longitudinal stresses. The listed references, Payne and Dongelmans (1997) and Cuffey and Paterson (2010), are too general to support that specific hypothesis. I am unaware of recent observations of faster-than-usual ice flow downstream of sticky spots or bumps. Do you have observations or modeling references that could support the ice speed increase? Otherwise, I suggest that the hypothesis is reframed so that the observed changes in bedforms downstream of bumps are due to elevated basal water pressure and reduced basal friction, not basal ice speed.

**The consideration for basal water pressure and reduced basal friction in synthesizing longer bedforms was added to the revised manuscript in line 27: "[...] topographic highs can increase strain heating and basal meltwater production, elevating basal meltwater pressure and reducing basal friction in the downstream environment (Payne and Dongelmans, 1997; Cuffey and Paterson, 2010)." Your comment was also referenced in lines 31- 32: "However, identifying which forms and scales of "bumps" across a glaciated landscape may increase, decrease, or not affect ice-flow velocity, basal water pressure, and basal friction is not well understood." "Ice-flow velocity" was kept in these lines due to the decrease in ice-flow velocity between the upstream and on-top of bump ice-bed interactions.**

**The only other reference to increased ice-flow velocity downstream of bumps was in the Results and Discussion section in line 119. This was removed to now read "Increased sediment availability and basal meltwater that results from the strain heating on top of the bump (Payne and Dongelmans, 1997), increases downstream sediment transport efficiency (McIntyre, 1985; Pohjola and Hedfors, 2003; Winsborrow et al., 2010b), resulting in the greatest number and most elongate bedforms, as well as the greatest proportion of bedforms with low surface relief downstream of bumps (Fig. 1A; Fig 2; Fig 4A, 4B)."**

2.      Figure 1B: The concentric circles in panel B are hard for me to tie to the statistics listed in the accompanying boxes. I would much prefer 2D plots containing the same information, which in my opinion, would make it easier to spot trends and variability.

**A box plot representing bump volume was added to Fig. 1B to address this comment. The text providing numerical values for these volumes was made more clear in the text boxes of this same figure.**

3.     Figure 4AB: Is it possible to remove the gray background in panels A and B? It makes it hard for me to discern the color differences between lines on my monitor.

**This adjustment was made to the final version of Fig 4.**

---

## Editor Decision (ED1)

**'Differential impact of isolated topographic bumps on ice sheet flow and subglacial processes' by McKenzie et al.**

The authors have done an excellent job of addressing all the points raised by the reviewer and editor in this second round of reviews. The impact of the article is significantly strengthened – the most recent edits provide additional detail on the methods and results and stronger justification for the points made in the discussion. The overall quality of the writing is good but there are a few places where the text could be a little clearer or the reader is left to fill in some of the details – try to make your writing as explicit as possible. A few suggestions to improve the clarity of the article are listed below. I leave it to the authors to make any edits that they see fit and am delighted to say that my decision is to 'publish subject to technical corrections' (the article will not undergo any further review by the editor). Thank you for choosing to publish your research in The Cryosphere!

Pippa Whitehouse (Editor)
* * *
Line 58: you have not previously used the terminology 'isolated topographic highs', so reference to '…these isolated topographic highs' is ambiguous

Line 63: awkward logic – the text essentially says that the bumps record the influence of the bumps…

Line 69: the implications of the final clause in this sentence are unclear, are you referring to the fact that all the landforms will have experienced roughly the same deglaciation and post-glacial landscape evolution processes?

Section 1.1: refer to figure 1 somewhere in the section that describes the geographic setting of the study area

Lines 77-78: 'the lack of ice streaming seen in-between singular bump topography… allowed us to treat all bumps as "aggregate" features' – terminology is a little unclear, perhaps clarify what you mean by 'singular bump topography'

Lines 105-110: the magnitude of the GIA signal is significant, but it will be spatially uniform across each region of interest. Conversely, any tectonics or post-glacial erosion will have a spatially variable signature, but the magnitude of the signal will be much smaller than the GIA signal. I think these observations can be used to justify the neglect of all three processes in your study but at the moment the argument is not very clear. For example, in the final sentence you refer to timescale discrepancies but then go on to talk about spatial resolution.

Line 117: be more explicit about what you mean by 'large timescales of uniform glaciation'

Line 134-135: it took me several goes to work out what you are trying to say in this sentence (I think you are saying that, statistically, bedforms on top of bumps have the lowest elongation and the greatest surface relief?) – suggest simplifying

Line 137: add a reference to figure 2 to support the statement about there being more bedforms downstream of a bump than upstream

Line 154: text missing? '…and *an increase in* surface relief…'

Line 161: decrease > increase?

Lines 166-167: 'increased pressure and basal drag' – wouldn't this increase the capacity for erosion?

Line 170: '…the similarities…are similar' – rephrase

Line 176: surface relief > surface area

Line 178: demonstrating > demonstrates

Line 181: could simplify, e.g. '…bumps with a volume greater than several cubic kilometres…'

Lines 182-183: '…bumps below this threshold cannot regain the same organization…' – needs a little more explanation, it is not the bumps that are organized…

Lines 183: 'This analysis found little evidence of channelized meltwater' – it is not clear which of your results supports this statement (is it actually a result from one of your other papers?)

Line 197: refer directly to your analysis, e.g. 'Overall, our results suggest…'

Line 201: 'The two largest bumps' and 'only bumps larger than several cubic kilometres' – do these essentially refer to the same thing? Can you streamline this sentence?

---

## Author Response (AR2)

**TC Review Response (Apr 2023)**

Dear authors,
Thank you for your submission to TC/TCD. As you may know, papers accepted for TCD appear immediately on the web for comment and review. Before publication in TCD, all papers undergo a rapid access review undertaken by the editor and/or reviewer with the aim of providing initial quality control. It is not a full review, and the key concerns are fit to the journal remit, basic quality issues and sufficient significance, originality and/or novelty to warrant publication. As a result, even a manuscript ranked highly during access review can receive a low ranking during full peer review later. Evaluation criteria are found at www.thecryosphere.net/review/ms_evaluation_criteria.html. Grades are from 1 (excellent) to 4 (poor). My evaluation is found below, and my recommendation is to publish the manuscript in TCD and proceed to the open discussion and peer-review. Thank you again for submitting your work to this journal.

With kind regards,
Pippa Whitehouse (Editor)

ORIGINALITY / NOVELTY (1-4): 2
This study presents an analysis of the morphometry of glacial landforms located upstream, downstream, and on top of bedrock topographic highs in the Puget Lowland. This is a novel piece of work, which employs GIS tools to quantify the properties of the landforms and it will potentially be of interest to readers of The Cryosphere. However, the interpretation of the results is currently somewhat limited.

SCIENTIFIC QUALITY / RIGOR (1-4): 3
 A significant amount of analysis has been carried out and the results are documented in a series of well-presented figures. However, the main findings are currently not strongly supported by the text. There is a relatively brief summary of the results and no discussion of the limitations of the study (How accurate is the automated tool? How important are feedbacks between ice flow and landform development?). There is also no appreciation of how local factors may influence the results. Statistical tests are used to identify robust differences in the means of populations, but the claim that a bump volume of 4.5 km$^3$ is the threshold for significantly impacting ice flow and subglacial processes is based on a limited sample size. In general, much stronger (quantified) evidence is needed to support some of the statements in the 'Results and Discussion' section. There is also a tendency to speculate about the processes operating, e.g. erosion/sedimentation, and the implications for ice flow, the role of basal meltwater etc. Arguments need to be developed more carefully and supported by evidence and the existing literature. Lastly, some aspects of the methods are unclear, and the language used

could be more specific. For example, what does it mean to 'assess' the bedrock bumps? Or determine the 'significance' of bedform characteristics?

**Consideration of local drivers of glaciation, in addition to topography, were added to the introduction in lines 50-54. Clarifications were added in the Methods section to address the accuracy of the automated tool (lines 91-92) and ambiguity of the terms "assess", and "significance" (lines 128-130) of bedform characteristics. While feedbacks between ice flow and landform development are thought to be very important, they are not well understood and negligible in this work due to the relative size of bumps to bedforms. This point was clarified in the text. To address concerns related to statements in 'Results and Discussion', the paper was reworked to include separate 'Results' and 'Discussion and Interpretation' sections, where the latter more carefully develops statements with support from outside literature.**

SIGNIFICANCE / IMPACT (1-4): 2/3
Since there is no discussion of how local factors may influence the results, it is difficult to assess how applicable the findings may be to other settings. There is also no specific guidance on how the findings could be used to inform future work. When considering the impact of the research, I encourage the authors to think about what they can directly say as a result of their analysis, before speculating about the processes operating or the impacts on ice flow.

**To address these concerns, an additional paragraph was added to the Introduction and new 'Discussion and Interpretation' section to consider local factors. Results were then separated from interpretations to develop a more clear understanding of direct data analysis.**

PRESENTATION QUALITY (1-4): 2
The manuscript is generally well organised and well written, but the text could be more specific/detailed when describing the aims of the study or documenting the methods and results. Check that all text has a purpose: taking an example from the abstract, review what you are seeking to communicate by the phrase 'not all bumps have the same degree of impact'. Figures are generally informative and of good quality.

**The phrase 'not all bumps have the same degree of impact' was clarified in the text to mean larger bumps cause greater disorganization of ice flow, while smaller bumps seem to have little to no effect on ice and subglacial processes (lines 22-23).**

**Thank you to the editor for all of these comments and suggestions – they strengthen the structure of the work and improve its quality through further developing the strength and significance of interpretations.**
* * *
Dear authors and editor,

I read with interest this well written manuscript that presents an interesting study on the relationship between ice sheet bedform metrics and bedrock "bumps", and how this possible link can be used to inform how bumps affect ice sheet dynamics. I was asked to look at this manuscript with fresh eyes and, while I see that the authors have done a very good job at addressing the small concerns of the previous reviewers, and while the approach and results are certainly robust, I find that there is room for further improvement. Specifically, I would argue that a little more description of the results can be provided, and that the interpretations require a little more effort. Arguably, figures can do the job of describing results, and a concise discussion can be good if the interpretations are straightforward. However, I feel that the manuscript is a bit of a missed opportunity if some of the complications and limitations of the study and the topic in general are not at least mentioned. I am worried that, without such effort, some of the conclusions might end up representing oversimplification of an otherwise complex set of environments and processes. Overall, I suggest the manuscript is accepted for publication but pending the inclusion of more details on results and, especially, a stronger discussion as per below.

Methodology:
1. What is the process-linked justification for considering the lateral (relative to the bumps) bedforms in the downstream group?

**Due to the possible influence of bumps on created lateral shear margins and thus lateral differentiation in ice flow speed, bedforms along the side of the bumps were considered to potentially have been impacted by the presence of the bumps: hence the classification "downstream" of the bump. This clarification was added to the text in lines 112-115.**

2. The relief is an interesting metric, but, if I have not misunderstood it, wouldn't its interpretation depend on how "quickly" a certain relief is attained and over what length? In other words, depending on where the max and min elevations are found, relative to the profile? How steep is the profile, and how consistently so along the long axis? For example, a 200 m long bedform with a 20 m relief could be linked to different processes than a 2000 m long bedform with the same relief. A bedform where the point of maximum elevation (say 20 m above the minimum elevation) is reached in the first (upstream) 10 m of its 2000 m long axis and another one where the same elevation is gained much slower (say the 20 m elevation gain

is reached halfway through the 2000 m long bedform) might be linked to different process(es). Perhaps this is all covered in your other, cited paper, but it should at least be briefly mentioned here too as it is rather important relative to the interpretations.

**The small-scale interpretations of surface relief change across a single bedform long-axis could be influenced from tectonics or post-glacial erosion and are not able to be teased-out from this dataset. The surface relief presented in this work considers larger-scale comparisons that generally point to regional processes of subglacial erosion, deposition, and deformation during bedform synthesis and do not capture the smaller scale post-glacial reworking of material. This point was clarified in the Methods.**

**Glacial isostatic adjustment occurs on a large spatial scale across the region, so does not influence the presentation of relative bedform surface relief whereas the smaller-scale tectonic activity and erosion may obscure the data if surface relief were considered on a smaller scale, such as across the long axis of a single bedform. This comparison in scale was clarified in the Methods.**

3. Some of the study sites have more than one bump. This should be mentioned and its potential effect on processes acknowledged/considered.

**Some of the sites have more coarse topography than others, but because no ice streaming (determined from the presence of streamlined subglacial bedforms) was seen in between a single-site topography, all sites were treated as "aggregate" bumps. This clarification was added in lines 76-78.**

Results:
4. More details could be provided. Specifically, what is the actual difference (e.g. on average/median) in the various metrics between bedforms up-stream of, on, and down-stream of the bumps?

**Data was added throughout the newly developed 'Results' section to address this comment.**

Interpretations/discussion:
5. From what I can see from the figures, the difference in metrics is considerable and consistent between bedforms on a bump and up- or down-stream ones. However, I seem to notice that the difference between up- and down-stream bedforms is often within the noise and certainly not consistent in the 9 cases considered. If so, this should be more clearly stated and considered as it might have implications on some of your interpretations. In other word, some

of the described processes used to justify the bedform metrics down-stream might not apply up-stream, and yet the metrics are similar/comparable.

**Thank you for sharing this point – despite similarities between upstream and downstream environments, the downstream bedforms indicate 'recovery' of the system from the impact of moving over the bump. Because the bump has a clear influence, the finding is referring to the ability of the system to once again recover to the same metrics and upstream of the bump despite different processes applying downstream that are not relevant upstream. There are also many more bedforms downstream of the bumps, even though the metrics between upstream and downstream are similar. This point was clarified in the 'Discussion and Interpretation' section.**

6. The interpretations are largely based on previous papers' attempt to link elongation, relief etc. to velocity and other ice sheet characteristics. Although these are published, I would argue that a little more details can be provided. For example, what do we mean by maturity or efficiency? What are the links between these processes? For example, is an efficient bed linked to high velocities? I would argue that it is important to provide some of these info and not give these for granted.

**Clarifications were added to address these points – in the context of this work, maturity refers to the persistence of warm-based ice flow and efficiency refers to how well distributed sedimentary processes are across the landscape.**

7. Some of the connections between bedform metrics and ice sheet characteristics, published in previous papers, might rely on a number of assumptions that should be at least acknowledged here. For example, I suspect that sediment starvation, porosity, bedrock fracturing etc. could all have had an impact in the metrics of bedforms. You say upfront that bedrock characteristics are out of scope, which I perfectly understand, but perhaps these are key for the interpretation of some of your results, so I would argue that they should form part of the discussion, even if you cannot reach a clear-cut conclusion. There are other aspects to consider too. What if the bedforms in each site are time-transgressive and reflect different phases (and possibly processes) of the ice sheet flow history? Can we resolve speed vs. duration when it comes to their influence on the metrics used here? Are the bedforms made up of sediment or bedrock? For example, some of the interpretations referred to in the manuscript seem more pertinent to bedrock bedforms than sedimentary ones (e.g. when you talk about erosion). I am not suggesting, of course, that you resolve all these assumptions/limitations, but it is important that they are at least mentioned/acknowledged.

**Within the same bump, persistent flow paths and similar rates of glacial isostatic adjustment uplift render the evolution of bedforms compared up- to downstream not relevant. While we can resolve relative speed and apparent spatial homogeneity of subglacial processes, whether elongation is an indication of duration or speed of ice cover cannot be directly resolved from this data. The variations in bedform composition are clarified in the 'Methods' and explanation for why sedimentary and erosional features are considered together was added to this same region (lines 119-124).**

8. I think most of the key papers used here for the interpretation on ice sheet dynamics come from palaeo studies. It would be interesting, and arguably important, to see what models and studies on present-day Antarctica and Greenland suggest with relation to these links and include a mention to these where pertinent.

**This comment was addressed by including an additional paragraph in the 'Discussion and Interpretation' section. Papers referenced that consider these applications include Holschuh et al., 2020, Eyles et al., 2018, and Greenwood et al., 2021.**

I know it is probably annoying to receive these comments at a stage where you felt the manuscript should be essentially accepted in its resubmitted form and I would perfectly understand if you and/or the Editor decide to ignore the comments above. However, I do believe that it will be a much stronger paper if you could consider at least some of the raised points and, where valid and useful, use these to improve the study.

With best wishes,
Matteo

**Thank you immensely for these comments – we agree that combined with the Editor's comments the work was greatly improved by clarifying much of the results and discussion. Your insight is very much appreciated.**
* * *
Notification to the authors:
The title page of *pdf. manuscript file must include the full institutional addresses of all authors. However, country name is missing from the affiliations. Please add it for the next revision.

**This edit was made to the manuscript.**